# Accelerating Diffusion LLM Inference via Local Determinism Propagation

## Abstract

Diffusion large language models (dLLMs) represent a significant advancement in text generation, offering parallel token decoding capabilities. However, existing open-source implementations suffer from quality-speed trade-offs that impede their practical deployment. Conservative sampling strategies typically decode only the most confident token per step to ensure quality (*i.e.*, greedy decoding), at the cost of inference efficiency due to repeated redundant refinement iterations–a phenomenon we term delayed decoding. Through systematic analysis of dLLM decoding dynamics, we characterize this delayed decoding behavior and propose a training-free adaptive parallel decoding strategy, named *Local-Leap*, to address these inefficiencies. *LocalLeap* is built on two fundamental empirical principles: local determinism propagation centered on high-confidence anchors and progressive spatial consistency decay. By applying these principles, *LocalLeap* identifies anchors and performs localized relaxed parallel decoding within bounded neighborhoods, achieving substantial inference step reduction through early commitment of already-determined tokens without compromising output quality. Comprehensive evaluation on various benchmarks demonstrates that *LocalLeap* achieves $6.94\times$ throughput improvements and reduces decoding steps to just 14.2% of the original requirement, achieving these gains with negligible performance impact. The source codes are available at: https://anonymous.4open.science/r/LocalLeap-Code-806C.

## 1 Introduction

Large language models (LLMs) have revolutionized natural language processing (NLP) by achieving leading performance across various tasks and becoming the dominant paradigm in the field (Zhao et al., 2023; Minaee et al., 2024). Mainstream autoregressive (AR) LLMs (Grattafiori et al., 2024; Yang et al., 2025a; Guo et al., 2025) rely on sequential next-token prediction, but this strict left-to-right serialization inherently limits throughput despite the approach's stability and implementation maturity. Recently, diffusion large language models (dLLMs) (Google DeepMind, 2025; Song et al., 2025b; Nie et al., 2025b) have gained considerable traction in the community. By leveraging bidirectional attention and iterative "mask-and-denoise" refinement, dLLMs enable parallel token decoding, delivering flexible and accelerated generation that establishes them as a promising paradigmatic shift.

Despite eliminating sequential dependencies, existing open-source dLLMs suffer from quality-speed trade-offs that limits their practical deployment, often exhibiting slower inference than AR LLMs. Common confidence-based sampling strategies (Chang et al., 2022) select a fixed number of high-confidence tokens at each step but typically decode only the single most confident token to ensure quality (*i.e.*, greedy decoding), resulting in inefficient inference (Ye et al., 2025). To examine whether such redundant refinements genuinely contribute to generation quality, we track and analyze the step-by-step decoding dynamics of dLLMs. We find that many token positions reach consistency with their final states early and remain stable throughout the prediction process, revealing that overly conservative sampling strategies unnecessarily delay determination and introduce redundant decoding steps–a phenomenon we term delayed decoding.

To alleviate this phenomenon, we first explore the prerequisites for effective parallel decoding. We find that the emergence of high-confidence tokens (termed anchors) produces a stabilizing effect in their vicinity: predictions for nearby tokens become more reliable and tend to remain consistent with

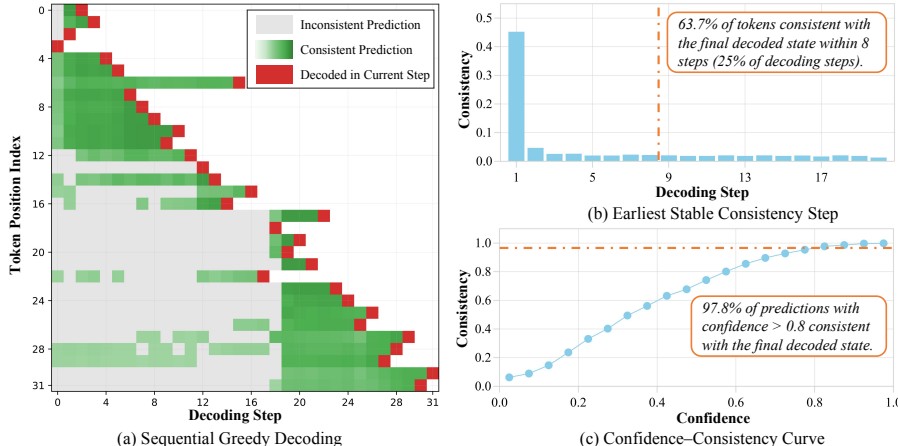

(a) Sequential Greedy Decoding

(b) Earliest Stable Consistency Step

(c) Confidence–Consistency Curve

Figure 1: **Confidence-aware sequential decoding visualization and analysis.** (a) Sequential greedy decoding visualization on a GSM8K instance using LLaDA-8B-Instruct. Gray: inconsistent predictions; Green: consistent predictions (intensity indicates confidence from 0 to 1); Red: decoded tokens. (b) Distribution of earliest stable consistency steps, showing 63.7% of tokens achieve final consistency within 25% of total decoding steps. (c) Confidence-consistency relationship demonstrating that 97.8% of predictions with confidence > 0.8 remain consistent with final outputs.

their final decoded states. This observation suggests that once anchors appear, surrounding regions can be decoded using less conservative rules, rather than requiring strict greedy commitment for each token. Based on this insight, we propose *LocalLeap*, a training-free adaptive parallel decoding strategy. Our method identifies anchors in the sequence and performs parallel decoding within a bounded radius around each anchor, using a relaxed confidence threshold to define the decoding boundary. This adaptive sampling strategy ensures the effective decoding of determinable tokens, effectively eliminating unnecessary refinement iterations. We make three key contributions:

- We characterize the behavioral dynamics of dLLMs during sequential greedy decoding, revealing a delayed decoding phenomenon. Based on this analysis, we formulate two empirical principles–local determinism propagation and spatial consistency decay–which establish the theoretical guidance for dLLM acceleration.
- We propose *LocalLeap*, a training-free adaptive parallel decoding strategy that identifies anchors and performs localized relaxed parallel decoding within bounded neighborhoods, achieving substantial inference step reduction without compromising output quality.
- Through extensive experiments across various benchmarks, we demonstrate that our proposed *LocalLeap* achieves 6.94× throughput improvements and significant reductions in inference steps, while preserving model performance with negligible degradation. For example, *LocalLeap* uses only 14.2% of inference steps while achieving slightly superior performance compared to greedy sequential decoding on GSM8K benchmark with LLaDA-Instruct (Nie et al., 2025b).

## 2 PRELIMINARY

### 2.1 INFERENCE AND DECODING STRATEGIES

Given a prompt $\mathbf{p} = (p_1, \ldots, p_K)$ and a target generation sequence length $L$, we define the initial state of response $\mathbf{r}$ as

$$\mathbf{r}_0 = \big( \underbrace{[\texttt{MASK}], \ldots, [\texttt{MASK}]}_{L \text{ tokens}} \big) \tag{1}$$

A dLLM can be viewed as a *mask predictor* that iteratively refines this noised sequence over steps $s = \{1, \ldots, T\}$. At step $s$, for each masked position $i$, the model $p_\theta$ produces a categorical distribution, from which we define the most likely token $\hat{r}_s^i$ and its confidence score $c_s^i$ by

$$\hat{r}_s^i = \arg\max_{v \in \mathcal{V}} p_\theta(r_s^i = v | \mathbf{p}, \mathbf{r}_{s-1}), \quad \text{and} \quad c_s^i = p_\theta(r_s^i = \hat{r}_s^i | \mathbf{p}, \mathbf{r}_{s-1}) \tag{2}$$

In theory, the model can unmask all masked tokens in one step. However, to ensure generation quality, dLLM employs a transition function $S$ to selectively commit a subset of the predictions $\hat{\mathbf{r}}_s$ when forming the next state $\mathbf{r}_s = \mathcal{T}(\hat{\mathbf{r}}_s, \mathbf{r}_{s-1})$. where transition function $\mathcal{T}$ refer to as *remasking* approch. In the following, we describe the most commonly used remasking schemes.

**Low-Confidence Remasking.** At each step $s$, the confidence scores $c_s^i$ for masked positions are given by Eq. 2. In this straightforward strategy, the predictor commits the unmasked positions with the highest confidence and leaves the remaining positions as [MASK].

**Semi-Autoregressive Remasking.** Although dLLMs do not enforce a strict left-to-right dependency and can update arbitrary positions, instruct models (trained on corpora containing abundant [EOS] tokens) often exhibit a tendency to emit [EOS] tokens prematurely. A practical remedy is to divide the sequence $\mathbf{r}$ into blocks and perform generation in a left-to-right block-wise fashion (Nie et al., 2025b). Specifically, during consecutive steps, the model restricts its decoding and remasking operations only within the current block.

**Confidence-Aware Parallel Decoding.** The bidirectional attention and arbitrary-position update capability of dLLMs make them inherently well-suited for parallel decoding, but conservative quality safeguards often limit decoding speed. Suppose a set of $n$ to be predicted positions given a conditioning context $\mathcal{E} = \{\mathbf{p}, \mathbf{r}_{s-1}\}$. Let $\mathbf{r}_s = (r_s^1, \ldots, r_s^n)$ denote the noised sequence, and let $\mathbf{r}^*$ be the particular sequence of interest. If $c_s^i$ are highly confident: $c_s^i > 1 - \epsilon$ with $\epsilon \leq \frac{1}{n+1}$, then greedy parallel decoding and greedy sequential decoding coincide (Wu et al., 2025): $\arg\max_{\mathbf{r}_s} p(\mathbf{r_s}|\mathcal{E}) = \arg\max_{\mathbf{r}_s} q(\mathbf{r}_s|\mathcal{E}) = \mathbf{r}_s^*$, where $p(\cdot|\mathcal{E})$ denotes the joint conditional probability mass function (PMF) and $q(\cdot|\mathcal{E}) = \prod_{i=1}^n p_i(\cdot|\mathcal{E})$ denotes the product-of-marginals approximation.

## 2.2 DELAYED DECODING

Despite the theoretical advantages of parallel decoding capabilities in dLLMs, conservative sampling schedules frequently underutilize bidirectional attention and parallel-update mechanisms. We hypothesize that existing sampling strategies induce numerous redundant refinement steps, leading to suboptimal inference efficiency.

To validate this hypothesis, we analyze the decoding behavior of LLaDA-8B-Instruct (Nie et al., 2025b) under both low-confidence remasking and semi-autoregressive remasking regimes. We track per-step predictions at each token position and compare them against the final decoded output. Our analysis employs two key concepts:

- *Consistency*: *Consistency*, where the prediction at the current step matches the final decoded token.
- *Stability*, where the prediction remains unchanged through subsequent refinement iterations.

Figure 1(a) illustrates the decoding process for a representative GSM8K question, demonstrating that numerous token positions generate predictions in early steps that already match the final greedy decode. Despite this early alignment, conservative remasking schedules inhibit premature commitment of these positions, deferring their resolution to later steps.

To quantify this delayed decoding phenomenon, we sample 120 examples (40 each from GSM8K, HumanEval, and IFEval) and record the earliest step where each position first becomes both consistent and stable. Our findings, presented in Figure 1(b), demonstrate that 45.2% of positions achieve consistency and stability at the first step within a block, while 63.7% could be effectively committed within the first 25% of decoding steps. This delayed commitment pattern significantly increases refinement iterations and overall inference cost.

While confidence serves as an effective decoding signal, strict top-1 or fixed-threshold policies compromise decoding efficiency. As shown in Figure 1(c), among predictions with confidence exceeding 0.8, at least 97.8% maintain consistency with the final decode. However, excessive threshold reduction introduces more erroneous predictions (Wu et al., 2025; Ben-Hamu et al., 2025) while improving efficiency, necessitating a balance between decoding speed and generation quality.

This analysis establishes a clear optimization objective: enable early commitment of reliably determined tokens to minimize total refinement steps. The central challenge becomes: *how can we reliably identify positions suitable for effective early commitment at each decoding step?*

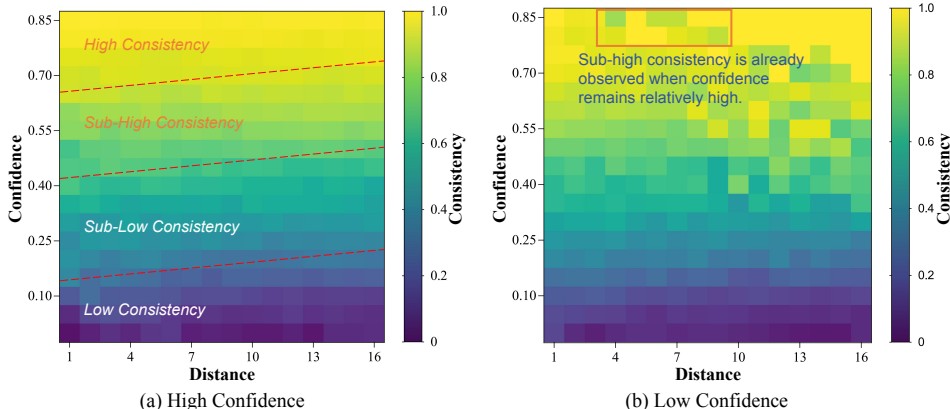

(a) High Confidence        (b) Low Confidence

Figure 2: **Heatmap of consistency analysis with confidence surrounding decoded tokens.** Centered on the decoded tokens at each step, we analyze the trends in confidence and consistency across surrounding positions. (a) When a decoded token exhibit high confidence ($c \geq 0.9$), its nearby positions maintain high consistency even at moderately lower confidence levels, while more distant positions require correspondingly higher confidence thresholds. (b) When decoding tokens exhibit low confidence ($c < 0.9$), consistency remains poor even at sub-high confidence levels ($c \in (0.8, 0.9)$), indicating that premature decoding at these confidence levels may introduce errors.

## 3 METHODOLOGY

We introduce *LocalLeap*, a training-free adaptive parallel decoding schedule for dLLMs. Our approach leverages high-confidence predictions as anchors to enable safe parallel decoding within localized neighborhoods, substantially reducing refinement steps while maintaining output quality. *LocalLeap* requires only a simple rule-based modification to existing sampling schedules, making it readily applicable to current dLLM implementations.

### 3.1 PREREQUISITE OF SAFE PARALLEL DECODING

To address the delayed decoding challenge identified in Section 2.2, we investigate this through systematic analysis of confidence and consistency patterns around decoded positions. Using the same 120 examples from our preliminary analysis (40 each from GSM8K, HumanEval, and IFEval), we examine the relationship between spatial proximity, confidence levels, and prediction reliability.

For each decoding step, we center a bidirectional sliding window on the token being committed and analyze surrounding masked positions within a maximum distance of 16 in both directions (±16 positions). This bidirectional analysis is motivated by dLLMs' bidirectional attention mechanisms, which enable tokens to influence and be influenced by context from both preceding and subsequent positions. We record two key metrics: (1) confidence scores $c$ at each position, and (2) consistency denoting whether current predictions match the corresponding tokens in the final decoded outputs.

Stratifying results by whether the central token exhibits high confidence ($c \geq 0.9$) or low confidence ($c < 0.9$), we generate the heatmaps shown in Figure 2. This analysis reveals two fundamental empirical principles that govern effective parallel decoding in dLLMs:

- *Local Determinism Propagation.* High-confidence anchor tokens create regions of enhanced predictive reliability in their immediate neighborhood. When the central token exhibits high confidence, neighboring positions with moderately high confidence (sub-threshold but substantial) demonstrate strong consistency with final outputs, enabling safe parallel commitment. Conversely, when the central token has low confidence, even moderately confident neighbors exhibit reduced consistency, making parallel decoding risky under such conditions.
- *Spatial Consistency Decay.* As demonstrated in Figure 2(a), consistency systematically decreases as distance increases, even for tokens within moderately high confidence ranges. Despite dLLMs' bidirectional attention mechanisms, the reliability of parallel commitments deteriorates with increasing spatial distance from high-confidence anchors. While high-confidence emergence enables

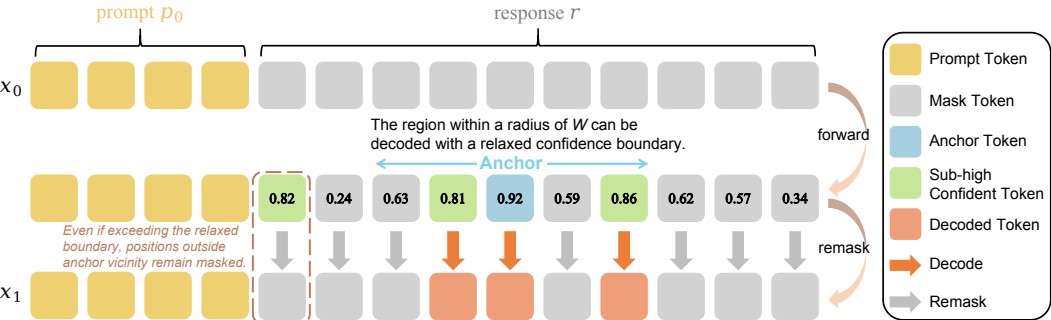

Figure 3: **Illustration of our LocalLeap decoding mechanism.** At each decoding step, we first compute confidence scores for all masked tokens through a forward pass, then identify anchors (tokens with confidence $c \geq 0.9$). We expand outward by $W$ positions from each anchor, creating local neighborhoods where tokens can be decoded using a relaxed confidence threshold $\tau = 0.75$. This allows certain tokens to bypass redundant optimization steps, thereby reducing the total number of decoding iterations.

relaxed decoding criteria within local neighborhoods, extending this relaxation indiscriminately across larger distances compromises decoding safety.

## 3.2 LOCALLEAP

Building upon these observations, we propose *LocalLeap*, a simple yet effective rule-based approach that systematically addresses the delayed decoding phenomenon in dLLMs. Specifically, *LocalLeap* operationalizes the empirical principles established in Section 3.1 through a systematic rule-based framework that identifies high-confidence anchor tokens and enables localized parallel decoding within their spatial neighborhoods using relaxed confidence boundary.

At decoding step $s$, let $\mathcal{B}_s$ be the active decoding block and $\mathcal{M}_s \subseteq \mathcal{B}_s$ the set of masked indices awaiting decoding. For each masked position $i \in \mathcal{M}_s$, mask predictor $p_\theta$ produces a confidence score $c_s^i$ as defined in Eq. 2. Our objective is to identify a subset $\mathcal{D}_s \subseteq \mathcal{M}_s$ that can be safely decoded in parallel at step $s$.

We first identify high-confidence tokens that can serve as reliable *anchors* for triggering localized parallel decoding:

$$\mathcal{A}_s = \{\, i \in \mathcal{M}_s \mid c_s^i \geq \kappa \,\}, \tag{3}$$

where $\kappa$ is a high-confidence trigger boundary. If $\mathcal{A}_s \neq \varnothing$, we enable local parallel decoding with a relaxed acceptance boundary within spatial neighborhood of radius $W$ around the anchors:

$$\mathcal{N}_s(\mathcal{A}_s) = \bigcup_{i \in \mathcal{A}_s} \{\, j \in \mathcal{M}_s \mid |j - i| \leq W \wedge j \neq i \,\}. \tag{4}$$

This neighborhood construction respects the spatial consistency decay principle by limiting the scope of relaxed decoding criteria to positions within bounded distance from anchors. The set of positions decoded at step $s$ combines anchors with qualifying neighbors:

$$\mathcal{D}_s = \mathcal{A}_s \cup \{\, j \in \mathcal{N}_s(\mathcal{A}_s) \mid c_s^j \geq \tau \,\}, \tag{5}$$

where $\tau < \kappa$ is a acceptance boundary for neighbors. Positions outside $\mathcal{N}_s(\mathcal{A}_s)$ continue to follow the conservative decoding rule. This strategy ensures that high-confidence anchors guide the parallel commitment of their sub-high confident neighbors, while maintaining reliability through distance-aware constraints. When no anchors are identified ($\mathcal{A}_s = \varnothing$), the algorithm returns to conservative sequential decoding, decoding only the single most confident position. This ensures robustness across diverse generation scenarios.

Algorithm 1 shows the decoding process of *LocalLeap*. We further establish its theoretical basis through formal analysis, including equivalence conditions with sequential decoding and distributional distance bounds, detailed in Appendix B. Notably, *LocalLeap* introduces negligible computational overhead beyond standard dLLM inference, requiring only confidence comparisons and simple

neighborhood checks. This makes *LocalLeap* readily deployable as a *plug-and-play* enhancement to existing dLLM implementations without retraining or architectural modifications.

---

**Algorithm 1** *LocalLeap*: Anchor-Guided Parallel Decoding

---

**Require:** model $p_\theta$, prompt $p_0$, generation length $L$, anchor trigger boundary $\kappa$, neighbor radius $W$, local relaxed boundary $\tau$

1: $x \leftarrow \text{concat}(p_0, \{\texttt{[MASK]}\}^L)$
2: **for** $s = 1$ to $S$ **do**
3:      $\mathcal{M} \leftarrow \{i \mid x^i = \texttt{[MASK]}\}$
4:      For masked $x^i$, confidence $c^i = \max_v p_\theta(x^i = v \mid x)$
5:      $\mathcal{A} \leftarrow \{i \in \mathcal{M} \mid c^i \geq \kappa\}$          ▷ Identify anchors
6:      **if** $|\mathcal{A}| > 0$ **then**          ▷ Parallel decoding
7:          $\mathcal{N} \leftarrow \text{GetNeighborSet}(\mathcal{M}, \mathcal{A}, W, \tau)$
8:          $\mathcal{D} \leftarrow \text{Union}(\mathcal{A}, \mathcal{N})$          ▷ Early decoding set
9:          **for** each $i \in \mathcal{D}$ **do**
10:             **if** $c^i \geq \tau$ **then**
11:               $x^i \leftarrow \arg\max_v p_\theta(x^i = v \mid x)$          ▷ Decode
12:             **end if**
13:          **end for**
14:      **else**          ▷ Fallback sequential decoding
15:          $i^* \leftarrow \arg\max_{i \in \mathcal{M}} c^i$
16:          $x^{i^*} \leftarrow \arg\max_v p_\theta(x^{i^*} = v \mid x)$          ▷ Decode
17:      **end if**
18: **end for**
19: **return** $x$

---

## 4 Experiments

### 4.1 Experiment setup

**Models and Baselines.** We conduct experiments on two representative open-source dLLMs: LLaDA-8B-Instruct (Nie et al., 2025b) and Dream-v0-Instruct-7B (Ye et al., 2025). For each model, we adopt semi-autoregressive remasking (Nie et al., 2025b) with block size 32 to prevent premature [EOS] token generation. We compare three decoding strategies:

- *Sequential Decoding.* Standard diffusion decoding, with only the most confident token decoded at each step.
- *Confidence-Aware Parallel Decoding* (Wu et al., 2025)**.** At each step, this method decodes all tokens with confidence above a threshold in parallel. If no tokens exceed the threshold, it falls back to decoding the single most confident token. The confidence threshold is set to 0.9.
- *LocalLeap.* Our anchor-guided localized parallel decoding approach. LocalLeap involves three hyperparameters: anchor trigger boundary $\kappa$, neighbor radius $W$, and local relaxed boundary $\tau$. Unless otherwise specified, we set $\kappa = 0.9$, $W = 4$, and $\tau = 0.75$ based on our preliminary experiments.

**Benchmarks and Metrics.** Our evaluation spans three domains: mathematics reasoning (GSM8K (Cobbe et al., 2021) and MATH (Hendrycks et al., 2021)), code generation (HumanEval (Chen et al., 2021) and MBPP (Austin et al., 2021c)), and alignment (IFEval (Zhou et al., 2023)). We measure decoding speed using throughput (tokens per second, TPS) and inference steps, and report TPS speedup and inference step reduction compared to the baseline.

All experiments are conducted on 8 NVIDIA H800 80GB GPUs. We employ greedy decoding without any random sampling strategies to ensure reproducible and comparable results.

### 4.2 Main Results

We report the performance and decoding efficiency of LLaDA-Instruct and Dream-Instruct across five benchmarks in Table 1 and Table 3, respectively. Overall, *LocalLeap* consistently improves decoding speed across all tasks and models. For LLaDA on the GSM8K benchmark, *LocalLeap* achieves a

Table 1: **Comprehensive benchmark results on LLaDA-8B-Instruct.** The performance and inference speed are compared with different decoding methods.

| Benchmark | Method | TPS | Speedup | Step | Reduction | Accuracy |
|---|---|---|---|---|---|---|
| GSM8K (5-shot) | LLaDA | 7.76 | 1.00× | 512 | 1.00× | 78.24 |
| | + Parallel (Fast-dLLM) | 41.22 | 5.31× | 95 | 5.39× | 78.24 |
| | + LocalLeap | 53.82 | 6.94× | 73 | 7.01× | 78.54 |
| MATH (4-shot) | LLaDA | 23.53 | 1.00× | 256 | 1.00× | 33.22 |
| | + Parallel (Fast-dLLM) | 60.52 | 2.57× | 97 | 2.64× | 32.98 |
| | + LocalLeap | 76.95 | 3.27× | 76 | 3.37× | 32.86 |
| HumanEval (0-shot) | LLaDA | 33.81 | 1.00× | 256 | 1.00× | 40.24 |
| | + Parallel (Fast-dLLM) | 103.48 | 3.06× | 77 | 3.32× | 40.85 |
| | + LocalLeap | 121.66 | 3.60× | 61 | 4.20× | 40.24 |
| MBPP (3-shot) | LLaDA | 15.64 | 1.00× | 256 | 1.00× | 29.40 |
| | + Parallel (Fast-dLLM) | 56.19 | 3.59× | 70 | 3.66× | 29.60 |
| | + LocalLeap | 71.37 | 4.56× | 58 | 4.41× | 30.60 |
| IFEval (0-shot) | LLaDA | 18.77 | 1.00× | 512 | 1.00× | 71.46 |
| | + Parallel (Fast-dLLM) | 44.97 | 2.40× | 206 | 2.49× | 71.10 |
| | + LocalLeap | 50.00 | 2.66× | 185 | 2.77× | 71.34 |

6.94× throughput improvement, representing an additional 1.63× speedup over confidence-aware parallel decoding from Fast-dLLM (Wu et al., 2025). Simultaneously, the inference steps are reduced by 7.01×, meaning that inference can be completed using only 14.2% of the original refinement steps. Notably, LocalLeap maintains acceptable accuracy fluctuations, with most cases showing less than one percentage point difference from the native method, and even providing superior performance in certain scenarios. For instance, on MBPP, LocalLeap delivers a 4.56× inference acceleration and 4.41× step reduction for LLaDA, while achieving a 1.2% performance improvement (30.6% vs. 29.4%). Similarly, for Dream, it provides a 4.28× inference acceleration and 4.41× step reduction, accompanied by a 1.2% performance improvement (55.2% vs. 54.0%).

These empirical results validate the effectiveness and reliability of *LocalLeap*, providing strong support for our hypothesis: the emergence of high-confidence anchors is often accompanied by high certainty in neighboring token clusters. Synchronously decoding anchor neighborhoods with relaxed confidence thresholds effectively reduces redundant refinement steps with negligible performance impact, thereby accelerating the decoding process. Furthermore, the consistent improvements across different model architectures (LLaDA and Dream) and task domains (mathematical reasoning, code generation, and instruction following) demonstrate that *LocalLeap* is a scalable and practical rule-based framework for *plug-and-play* dLLM inference acceleration.

**Scalability.** To examine the scalability of our method, we conduct experiments on longer generation sequences (length 1024). When the generation length is set to 1024, *LocalLeap* remains effective. Specifically, on GSM8K, *LocalLeap* achieves 11.95× throughput improvement and 12.05× fewer inference steps, with slightly better performance. On HumanEval and IFEval, it delivers consistent gains in throughput and step reduction, with performance fluctuations within 2 points.

Table 2: **Results of LLaDA-8B-Instruct on long sequence (1024 Tokens).**

| Benchmark | Method | TPS | | Step | | Acc. |
|---|---|---|---|---|---|---|
| GSM8K (5-shot) | LLaDA | 2.66 | 1.00× | 1024 | 1.00× | 77.26 |
| | + LocalLeap | 31.79 | 11.95× | 85 | 12.05× | 78.17 |
| HumanEval (0-shot) | LLaDA | 9.86 | 1.00× | 1024 | 1.00× | 45.12 |
| | + LocalLeap | 63.46 | 6.44× | 152 | 6.74× | 43.29 |
| IFEval (0-shot) | LLaDA | 7.93 | 1.00× | 1024 | 1.00× | 69.66 |
| | + LocalLeap | 36.93 | 4.66× | 215 | 4.76× | 69.54 |

### 4.3 ABLATIONS AND ANALYSIS

Our LocalLeap involves three core hyperparameters: anchor trigger boundary $\kappa$, neighbor radius $W$, and local relaxed boundary $\tau$. To explore optimal settings that achieve better quality-speed trade-offs, we conduct univariate analysis for each parameter, as shown in Figure 4. We analyze how these parameters affect inference performance and throughput. Our objective is to maximize

Table 3: **Comprehensive benchmark results on Dream-7B-Instruct.** The performance and inference speed are compared with different decoding methods.

| Benchmark | Method | TPS | Speedup | Step | Reduction | Accuracy |
|---|---|---|---|---|---|---|
| GSM8K (5-shot) | Dream | 4.48 | 1.00× | 512 | 1.00× | 73.54 |
| | + Parallel (Fast-dLLM) | 28.27 | 6.31× | 81 | 6.32× | 74.75 |
| | + LocalLeap | 31.01 | 6.92× | 73 | 7.01× | 72.02 |
| MATH (4-shot) | Dream | 27.91 | 1.00× | 256 | 1.00× | 42.98 |
| | + Parallel (Fast-dLLM) | 57.88 | 2.07× | 119 | 2.15× | 43.48 |
| | + LocalLeap | 65.32 | 2.34× | 103 | 2.49× | 43.20 |
| HumanEval (0-shot) | Dream | 29.89 | 1.00× | 256 | 1.00× | 55.49 |
| | + Parallel (Fast-dLLM) | 67.02 | 2.24× | 97 | 2.64× | 59.76 |
| | + LocalLeap | 73.18 | 2.45× | 85 | 3.01× | 57.93 |
| MBPP (3-shot) | Dream | 13.55 | 1.00× | 256 | 1.00× | 54.00 |
| | + Parallel (Fast-dLLM) | 56.30 | 4.15× | 69 | 3.71× | 55.60 |
| | + LocalLeap | 58.03 | 4.28× | 58 | 4.41× | 55.20 |
| IFEval (0-shot) | Dream | 18.44 | 1.00× | 512 | 1.00× | 48.68 |
| | + Parallel (Fast-dLLM) | 50.21 | 2.72× | 178 | 2.88× | 49.76 |
| | + LocalLeap | 52.79 | 2.86× | 165 | 3.10× | 49.88 |

inference speed while maintaining negligible performance impact. Specifically, we experiment using LLaDA-Instruct on HumanEval with maximum generation length of 256, block size of 32, and default hyperparameter settings $\{\kappa = 0.9, W = 4, \tau = 0.75\}$, varying only one parameter per experiment.

**Anchor Trigger Boundary.** The emergence of anchor tokens with $c > \kappa$ determines when to trigger LocalLeap decoding. Higher $\kappa$ implies stricter parallel triggering conditions; in the extreme case when $\kappa = 1$, it approximates sequential decoding. When $\kappa = 0.95$, we achieve $3.5\times$ throughput improvement; when $\kappa = 0.9$, throughput further increases to $4.1\times$ without performance degradation. However, as $\kappa$ further decreasing, performance drops significantly while throughput gains become marginal. This indicates that anchors require sufficiently high confidence (*e.g.*, $\kappa \geq 0.9$). As analyzed in Figure 2, treating lower-confidence tokens as anchors is unsafe, potentially decoding inconsistent tokens and degrading performance.

**Neighbor Radius.** Due to dLLMs' spatial consistency decay, anchor influence is spatially bounded. We quantify the quality-speed trade-off across different neighbor radiuses to determine the upper bound for safe decoding. When $W \in [1, 4]$, performance remains on par with or exceeds the baseline. When $W > 4$, performance degrades noticeably. In extreme $W = 32$, the method degenerates to block-wide parallel decoding with boundary $\tau$, and the significant performance drop reflects the necessity of parallel decoding locality. Moreover, throughput improvements plateau when $W \geq 4$. Overall, $W = 4$ achieves optimal quality-speed balance.

**Local Relaxed Boundary.** While dLLMs exhibit local determinism propagation, this does not imply all predictions in anchors' limited neighborhoods have converged. As shown in Figure 2(a), even in nearest regions, low-confidence positions maintain low consistency. Therefore, adopting a relaxed boundary is necessary. We investigate the most permissive conditions. When $\tau \in [0.65, 0.9]$, throughput improves significantly, indicating that anchor neighborhoods typically contain moderately high-confidence tokens. However, when $\tau < 0.75$, performance drops substantially, suggesting that 0.75 is the most relaxed yet safe decoding threshold. Additionally, different models exhibit varying confidence-consistency distributions (different sensitivity to $\tau$). For Dream-Instruct, 0.75 remains unsafe, so we slightly increase $\tau$ to 0.8.

## 5   RELATED WORK

**Diffusion Large Language Models.** Diffusion models (Sohl-Dickstein et al., 2015; Ho et al., 2020; Song et al., 2021), as a generative paradigm, have become the state-of-the-art approach for generating images (Nichol et al., 2022; Rombach et al., 2022) and videos (Ho et al., 2022; Blattmann et al.,

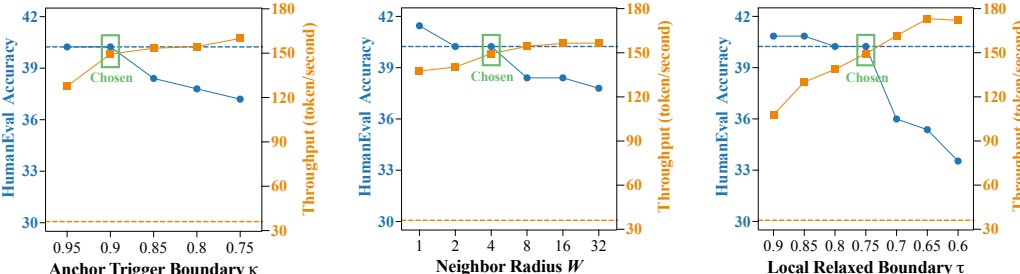

Figure 4: **Ablation study on hyperparameters:** anchor trigger boundary $\kappa$, neighbor radius $W$ and local relaxed boundary $\tau$ for LLaDA-Instruct on HumanEval. The default setting is $\{\kappa = 0.9, W = 4, \tau = 0.75\}$, and we perform univariate analysis for each variable. The blue line indicates accuracy, while the orange line indicates throughput. Each dashed line represents baseline performance for metrics of the same color.

2023). Recently, the emergence of discrete diffusion models (Austin et al., 2021a; Gong et al., 2025) has reshaped the landscape of text generation. Recent commercial models, such as Gemini Diffusion (Google DeepMind, 2025), Seed Diffusion (Song et al., 2025b) and Mercury (Labs et al., 2025), achieve remarkable inference speeds ranging from over 1,000 to more than 2,000 tokens per second. Recent open-source implementations, notably LLaDA (Nie et al., 2025b) and Dream (Ye et al., 2025), demonstrate promising results that match the performance of AR LLMs. Building upon these foundations, MMaDA (Yang et al., 2025b), LLaDA-V (You et al., 2025), and LaViDa (Li et al., 2025b) explore unified multimodal dLLM frameworks.

**Acceleration Methods for dLLMs.** Open-source dLLMs typically demonstrate limited inference efficiency, primarily due to computationally intensive iterative denoising and overly conservative sampling strategies (Chang et al., 2022). This limitation has prompted significant research efforts toward dLLM inference acceleration. Current acceleration methods can be classified two main categories. The first category utilizes KV caching mechanisms to mitigate edundant computations and memory consumption. For example, dLLM-Cache (Liu et al., 2025) combines long-interval prompt caching with partial response updates guided by feature similarity. dKV-Cache (Ma et al., 2025) proposes a delayed and conditioned caching strategy. Fast-dLLM (Wu et al., 2025) introduces a block-wise bidirectional KV cache mechanism. Sparse-dLLM (Song et al., 2025a) investigates dynamic cache eviction with sparse attention. The second category investigates optimized sampling strategies to reduce denoising iterations. For example, Fast-dLLM (Wu et al., 2025) and EB-Sampler (Ben-Hamu et al., 2025) perform dynamic parallel decoding based on confidence and entropy boundary conditions, respectively. However, overly strict threshold conditions force even correct tokens to meet unnecessary stringent requirements, thereby reducing overall efficiency. Recently, Prophet (Li et al., 2025a) discovers that dLLMs can terminate the entire decoding process early after sufficient iterations, reducing decoding steps at the sequence level. Unlike existing approaches, our work addresses token-level redundant optimization that occurs at specific positions during sequential inference, resulting in improved throughput and faster inference speeds.

## 6 CONCLUSION

In this work, we address the fundamental quality-speed trade-off limiting the practical deployment of dLLMs through systematic analysis of their decoding dynamics. Our investigation reveals delayed decoding phenomena, where conservative sampling strategies prevent early commitment of already-determined tokens, leading to substantial inference inefficiency. By establishing two key empirical principles of local determinism propagation around high-confidence anchors and spatial consistency decay, we develop *LocalLeap*, a training-free adaptive parallel decoding strategy that enables safe early decoding within anchor neighborhoods. Comprehensive evaluation across mathematical reasoning, code generation, and instruction-following benchmarks demonstrates that *LocalLeap* achieves up to $6.94\times$ throughput improvements and reduces inference steps to as low as 14.2% of the original requirements, both with negligible performance impact. These results establish *LocalLeap* as an effective solution to the dLLM efficiency bottleneck, providing a practical framework for accelerated diffusion-based text generation.

ETHICS STATEMENT

Our research focuses on algorithmic contributions to accelerate diffusion large language model (dLLM) inference. We have not collected any human subjects, private information, or sensitive data. All datasets used in our experiments are publicly available benchmarks (GSM8K, MATH, HumanEval, MBPP, and IFEval), and we strictly follow their usage licenses. The proposed method is training-free, rule-based, and does not require additional data collection or fine-tuning. This method does not pose foreseeable risks of misuse or societal harm beyond those already inherent to LLM.

REPRODUCIBILITY STATEMENT

We have made efforts to ensure the reproducibility of our work. Detailed experimental setups and hyperparameter configurations are provided in Section 4 and Appendix C. Notably, we employ greedy decoding for all baselines and our proposed method, without any random sampling or temperature settings, ensuring that model performance is deterministic and reproducible. To further support the validity of our approach, we include theoretical assumptions and complete proofs in Appendix B. To facilitate reproducibility, we provide an anonymous code repository along with scripts for training and evaluation.

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

## A    MORE PRELIMINARY

### A.1    FUNDAMENTALS OF DIFFUSION LARGE LANGUAGE MODELS

**Forward Masking Process.** Given a clean sample $x_0$ of length $N$, diffusion large language models (dLLM) forward process progressively replaces tokens with a special token [MASK] as a function of time $t \sim \mathcal{U}(0, 1)$ (Nie et al., 2025a). At time $t = 0$, the data point $x_0$ is fully observed with no masks, while at $t = 1$, all tokens are guaranteed to be masked. For a given $t \in (0, 1)$, the sequence $x_t$ is partially masked: each token $x_t^i$ is masked with probability $t$ and remains with probability $1 - t$. The transition probability for the noised sequence $x_t$ is

$$q_{t|0}(x_t \mid x_0) = \prod_{i=1}^{N} q_{t|0}(x_t^i \mid x_0^i), \quad \text{where } q_{t|0}(x_t^i \mid x_0^i) = \begin{cases} 1 - t, & x_t^i = x_0^i, \\ t, & x_t^i = \text{[MASK]}. \end{cases}$$

Here, M denotes the special mask token [MASK].

**Reverse Unmasking Process.** The forward process is reversible: the corresponding reverse process iteratively unmasks tokens, progressively evolving from $t = 1$ back to $t = 0$ (Austin et al., 2021b). Specifically, while masked positions either retain their masked state with some probability or undergo decoding to concrete tokens based on the model's predictive distribution. However, directly reversing Eq. A.1 is inefficient in practice, since it generally unmasks just one token per. A common remedy is the $\tau$-*leaping* approximation (Gillespie, 2001), which allows transitions from time $t$ to an earlier $s$ $(0 \le s < t \le 1)$ that simultaneously unmask multiple tokens.

$$q_{s|t} = \prod_{i=1}^{N} q_{s|t}(x_s^i \mid x_t), \quad \text{where } q_{s|t}(x_s^i \mid x_t) = \begin{cases} 1, & x_t^i \neq \text{M}, \ x_s^i = x_t^i, \\ \frac{s}{t}, & x_t^i = \text{M}, \ x_s^i = \text{M}, \\ \frac{t-s}{t} q_{0|t}(x_s^i \mid x_t), & x_t^i = \text{M}, \ x_s^i \neq \text{M}, \end{cases}$$

where $q_{0|t}(\cdot \mid x_t)$ is a categorical distribution over vocabulary $\mathcal{V}$ that predicts a non-[MASK] token at a masked position given the visible context $x_t$.

## B    THEORETICAL ANALYSIS

We establish theoretical guarantees for our LocalLeap parallel decoding scheme.

### B.1    DEFINITION

Let $\mathcal{D} = \{i_1, \ldots, i_n\}$ denote the set of masked positions to be decoded. For each $j \in \mathcal{N}$, and for some small $\epsilon > 0$, define

$$p_j^* = p_j(X_{i_j} = x_{i_j}^* \mid \mathcal{E}) > 1 - \epsilon,$$

be the model-assigned probability of the candidate token $x_{i_j}^*$ under context $\mathcal{E}$. We partition $\mathcal{N}$ into disjoint clusters $\mathcal{C}_1, \ldots, \mathcal{C}_K$. Each cluster $\mathcal{C}$ contains exactly one *anchor* position $a \in \mathcal{A}$ with high confidence $p_a^* \ge \kappa$, and its local neighborhood $\mathcal{N} = \{j \in \mathcal{D} \mid |j - a| \le W \wedge j \neq a\}$, where each $j \in \mathcal{N}$ satisfies $p_j^* \ge \tau$. We denote the number of neighbors in $\mathcal{N}$ by $m = |\mathcal{N}| \ge 1$. We assume thresholds satisfy $1/2 < \tau < \kappa < 1$. Define per-token error parameters

$$\epsilon_a < 1 - \kappa, \quad \epsilon_n < 1 - \tau, \quad \epsilon_{\max} = \max(\epsilon_a, \epsilon_n) < 1 - \tau,$$

and the aggregate error budget in the cluster

$$S = \sum_{j \in \mathcal{C}} p(X_{i_j} \neq x_{i_j}^* \mid \mathcal{E}) = \epsilon_a + m\epsilon_n < (1 - \kappa) + m(1 - \tau).$$

### B.2    EQUIVALENCE FOR SEQUENTIAL DECODING

$x^*$ **is the maximizer uniquely of** $q(x)$**.** Define the product-of-marginals probability mass function (PMF):

$$q(\boldsymbol{z} \mid \mathcal{E}) = \prod_{j=1}^{n} p_j(X_{i_j} = z_j \mid \mathcal{E}).$$

Maximizing $q(\boldsymbol{z} \mid \mathcal{E})$ factorizes into maximizing each marginal $p_j(X_{i_j} = z_j \mid \mathcal{E})$ independently. Since $p_j^* \geq \tau > 1/2$, each $x_{i_j}^*$ is the unique maximizer of its marginal distribution. Hence,

$$\arg\max_{\boldsymbol{z}} q(\boldsymbol{z} \mid \mathcal{E}) = (x_{i_1}^*, \ldots, x_{i_n}^*) = \boldsymbol{x}^*.$$

**Condition for $\boldsymbol{x}^*$ to maximize $p(x)$.** For a cluster $\mathcal{C}$ with $1 + m$ elements, the joint probability of the configuration $\boldsymbol{x}^*$ can be lower bounded by Bonferroni's inequality:

$$p(\boldsymbol{x}^* \mid \mathcal{E}) = p\Big( \bigcap_{j \in \mathcal{C}} \{X_{i_j} = x_{i_j}^*\} \Big| \mathcal{E} \Big)$$

$$\geq 1 - \sum_{j \in \mathcal{C}} p(X_{i_j} \neq x_{i_j}^* \mid \mathcal{E})$$

$$> 1 - \big( (1 - \kappa) + m(1 - \tau) \big).$$

For any alternative sequence $\boldsymbol{y} = (y_1, \ldots, y_n) \neq \boldsymbol{x}^*$, there exists at least one coordinate $k \in \mathcal{C}$ with $y_k \neq x_{i_j}^*$. For such $k$,

$$p(X_{i_j} = y_j \mid \mathcal{E}) \leq p(X_{i_j} \neq x_{i_j}^* \mid \mathcal{E}) \leq \epsilon_{\max} = 1 - \tau.$$

Thus,

$$p(\boldsymbol{y} \mid \mathcal{E}) \leq 1 - \tau.$$

To guarantee $\boldsymbol{x}^*$ is strictly more probable than any alternative, it suffices that

$$1 - \big( (1 - \kappa) + m(1 - \tau) \big) > 1 - \tau,$$

which simplifies to the cluster-level certification condition:

$$(1 - \kappa) + m(1 - \tau) < \tau.$$

When each cluster $\mathcal{C}$ satisfies the above inequality, $\boldsymbol{x}^*$ is the unique maximizer of $p(x)$ restricted to $\mathcal{C}$. Since clusters are disjoint, the global maximizer under $p(x)$ is the concatenation of all cluster-level maximizers:

$$\arg\max_{\boldsymbol{z}} p(\boldsymbol{z} \mid \mathcal{E}) = \arg\max_{\boldsymbol{z}} q(\boldsymbol{z} \mid \mathcal{E}) = \boldsymbol{x}^*.$$

**Neighbor Capacity Bound.** For a cluster $\mathcal{C}$ containing one anchor and $m$ neighbors, the certification condition

$$(1 - \kappa) + m(1 - \tau) < \tau$$

is equivalent to an explicit upper bound on the number of neighbors:

$$m < \frac{\tau - (1 - \kappa)}{1 - \tau} = \frac{\kappa}{1 - \tau} - 1.$$

Hence, the maximal admissible number of neighbors per anchor is

$$m_{\max} = \left\lfloor \frac{\kappa}{1 - \tau} - 1 \right\rfloor.$$

This bound guarantees that parallel commitments within each cluster are consistent with the sequential greedy decoding.

## B.3 $L_p$ DISTANCE IS BOUNDED

For discrete distributions $p$ and $q$, we define the $L_p$ distance as:

$$\|p - q\|_p^p = |p(\boldsymbol{x}^* \mid \mathcal{E}) - q(\boldsymbol{x}^* \mid \mathcal{E})|^p + \sum_{\boldsymbol{z} \neq \boldsymbol{x}^*} |p(\boldsymbol{z} \mid \mathcal{E}) - q(\boldsymbol{z} \mid \mathcal{E})|^p.$$

Let $A_j = \{X_{i_j} = x_{i_j}^*\}$ for $j \in \mathcal{C}$. By definition, we have:

$$p^* = p(\boldsymbol{x}^* \mid \mathcal{E}) = p\Big( \bigcap_{j \in \mathcal{C}} A_j \Big), \quad q^* = q(\boldsymbol{x}^* \mid \mathcal{E}) = \prod_{j \in \mathcal{C}} p(A_j).$$

**Bounds for $p^*$.** We first establish upper and lower bounds for $p^*$. By Boole's inequality:

$$p^* = p\Big(\bigcap_{j\in\mathcal{C}} A_j\}\Big) = 1 - p\Big(\bigcup_{j\in\mathcal{C}} A_j^c\Big) \geq 1 - \sum_{j\in\mathcal{C}} p(A_j^c) \geq 1 - \sum_{j\in\mathcal{C}} \epsilon_j = 1 - S.$$

where $S = \epsilon_a + m\epsilon_n$ is the aggregate error budget from the previous section.

For the upper bound, we have:

$$p^* = P\Big(\bigcap_j A_j\Big) \leq \min_{j\in\mathcal{C}} P(A_j) = 1 - \epsilon_{\max}.$$

Therefore,

$$1 - S \leq p^* \leq 1 - \epsilon_{\max}.$$

**Bounds for $q^*$.** For the product-of-marginals distribution, by Weierstrass's product inequality:

$$q^* = \prod_{j\in\mathcal{C}} p(A_j) = \prod_{j\in\mathcal{C}} (1 - \epsilon_j) \geq 1 - \sum_{j=1}^{n} \epsilon_j = 1 - S.$$

The upper bound follows directly:

$$q^* = \prod_{j\in\mathcal{C}} p(A_j) = \prod_{j\in\mathcal{C}} (1 - \epsilon_j) \leq \min_j (1 - \epsilon_j) = 1 - \epsilon_{\max}.$$

Therefore,

$$1 - S \leq q^* \leq 1 - \epsilon_{\max}.$$

**Bounding $|p^* - q^*|$.** Since both $p^*$ and $q^*$ lie in the interval $[1 - S, 1 - \epsilon_{\max}]$, we obtain:

$$|p^* - q^*| \leq (1 - \epsilon_{\max}) - (1 - \sum_{j=1}^{n} \epsilon_j) = \sum_{j=1}^{n} \epsilon_j - \epsilon_{\max} = S - \epsilon_{\max}$$

$$\leq (1 - \kappa) + (m-1)(1 - \tau)$$

**Bounds for alternative sequences.** For any $z \neq x^*$, we have:

$$p(z \mid \mathcal{E}) \leq \epsilon_{\max}, \quad q(z \mid \mathcal{E}) \leq \epsilon_{\max}.$$

Thus,

$$\sup_{z \neq x^*} |p(z \mid \mathcal{E}) - q(z \mid \mathcal{E})| \leq \epsilon_{\max}.$$

On the other side,

$$\sum_{z \neq x^*} |p(z \mid \mathcal{E}) - q(z \mid \mathcal{E})| = \sum_{z \neq x^*} \big(p(z \mid \mathcal{E}) + q(z \mid \mathcal{E})\big) = (1 - p^*) + (1 - q^*)$$

$$\leq \big(1 - (1 - S)\big) + \big(1 - (1 - S)\big) = 2S$$

Using the supremum bound: for any sequence $\{a_i\}$ with $|a_i| \leq M$ and $p \geq 1$, we have $|a_i|^p \leq M^{p-1}|a_i|$. So,

$$\sum_{z \neq x^*} |p(z \mid \mathcal{E}) - q(z \mid \mathcal{E})|^p \leq \big(\sup_{z \neq x^*} |p(z \mid \mathcal{E}) - q(z \mid \mathcal{E})|\big)^{p-1} \sum_{z \neq x^*} |p(z \mid \mathcal{E}) - q(z \mid \mathcal{E})|$$

$$\leq \epsilon_{\max}^{p-1} \cdot 2S$$

**Final $L_p$ distance bound.** Combining the above results:

$$\|p - q\|_p^p = |p(x^* \mid \mathcal{E}) - q(x^* \mid \mathcal{E})|^p + \sum_{z \neq x^*} |p(z \mid \mathcal{E}) - q(z \mid \mathcal{E})|^p$$

$$\leq (S - \epsilon_{\max})^p + (\epsilon_{\max}^{p-1} \cdot 2S)^p$$

**Total variation distance.** For the special case $p = 1$, the $L_1$ distance is:

$$\|p - q\|_1 \leq (S - \epsilon_{\max}) + 2S = 3S - \epsilon_{\max}$$
$$< 3(1 - \kappa) + (3m - 1)(1 - \tau)$$

Therefore, the total variation distance is bounded by:

$$\mathrm{TV}(p, q) = \frac{1}{2}\|p - q\|_1 < \frac{1}{2}\big(3(1 - \kappa) + (3m - 1)(1 - \tau)\big)$$

### B.4 KL DIVERGENCE IS BOUNDED

The total correlation between the joint distribution $p$ and the product-of-marginals $q$ is defined as the KL divergence, which can be expressed as the difference between the sum of marginal entropies and the joint entropy:

$$\text{TC}(p, q) = D_{KL}(p\|q) = \sum_{j \in \mathcal{C}} H(X_{i_j} \mid \mathcal{E}) - H(\boldsymbol{X} \mid \mathcal{E})$$

By the entropy chain rule:

$$H(\boldsymbol{X} \mid \mathcal{E}) = \sum_{k=1}^{m+1} H(X_{i_k} \mid X_{i_1}, \ldots, X_{i_{k-1}}, \mathcal{E}).$$

Substituting the chain rule decomposition:

$$\begin{aligned}
\text{TC} &= \sum_{k=1}^{m+1} H(X_{i_k} \mid \mathcal{E}) - H(\boldsymbol{X} \mid \mathcal{E}) \\
&= \sum_{k=1}^{m+1} \Big[ H(X_{i_k} \mid \mathcal{E}) - H(X_{i_k} \mid X_{i_1}, \ldots, X_{i_{k-1}}, \mathcal{E}) \Big] \\
&= \sum_{k=2}^{m+1} I(X_{i_k}; X_{i_1}, \ldots, X_{i_{k-1}} \mid \mathcal{E}),
\end{aligned}$$

where $I(X; Y \mid \mathcal{E})$ denotes conditional mutual information, and we use the convention that $I(X_{i_1}; \emptyset \mid \mathcal{E}) = 0$.

For all $k \geq 2$, conditional mutual information is bounded by marginal entropy:

$$I(X_{i_k}; X_{<k} \mid \mathcal{E}) = H(X_{i_k} \mid \mathcal{E}) - H(X_{i_k} \mid X_{<k}, \mathcal{E}) \leq H(X_{i_k} \mid \mathcal{E}),$$

Thus,

$$\text{TC} \leq \sum_{k=2}^{m+1} H(X_{i_k} \mid \mathcal{E}).$$

For any position $k \in \mathcal{C}$ with top-1 probability $p_k^* = p_k(X_{i_k} \mid \mathcal{E})$ and error rate $\epsilon_k = p(X_{i_k} \neq x_{i_k}^* \mid \mathcal{E}) \leq 1 - p_k^*$, the conditional entropy is bounded by:

$$H(X_{i_k} \mid \mathcal{E}) \leq H_b(\epsilon_k) + \epsilon_k \ln(|\mathcal{V}| - 1),$$

where $H_b(\epsilon) = -\epsilon \ln \epsilon - (1 - \epsilon) \ln(1 - \epsilon)$ is the binary entropy function, and $|\mathcal{V}|$ is the vocabulary size.

The entropy is maximized when the remaining probability mass $\epsilon_k$ is distributed uniformly over the $|\mathcal{V}| - 1$ non-optimal tokens.

Let the anchor have error rate $\epsilon_a < 1 - \kappa$ and each of the $m$ neighbors have error rate $\epsilon_n < 1 - \tau$. Then,

$$\begin{aligned}
\text{TC} &\leq H(X_{\text{anchor}} \mid \mathcal{E}) + \sum_{\text{neighbor } j} H(X_j \mid \mathcal{E}) \\
&\leq \Big[ H_b(\epsilon_a) + \epsilon_a \ln(|\mathcal{V}| - 1) \Big] + m \cdot \Big[ H_b(\epsilon_n) + \epsilon_n \ln(|\mathcal{V}| - 1) \Big] \\
&< \Big[ H_b(1 - \kappa) + (1 - \kappa) \ln(|\mathcal{V}| - 1) \Big] + m \cdot \Big[ H_b(1 - \tau) + (1 - \tau) \ln(|\mathcal{V}| - 1) \Big].
\end{aligned}$$

### B.5 PRACTICAL RELAXATION

The conditions and bounds established above are *worst-case*, hence sufficient but not necessary. In practice, several benign effects allow us to relax the certification condition and yield tighter realized bounds, without invalidating the theoretical guarantees.

The preceding conditions and bounds are *worst-case*, i.e., sufficient but not necessary. In practice, several benign effects allow us to relax them without violating consistency. First, anchor confidence typically exceeds the trigger $\kappa$, and neighbor confidences often lie well above the acceptance floor $\tau$. Equivalently, the realized error rates satisfy $\epsilon_a \ll 1 - \kappa$ and $\epsilon_n \ll 1 - \tau$. Substituting these realized rates into the cluster certificates (e.g., $(1-\kappa)+m(1-\tau) < \tau$) tightens both the equivalence condition and the distributional bounds ($L_p$ and KL), yielding strictly smaller margins on the left-hand sides and thus strictly smaller risk budgets. Second, the effective neighborhood size at a decoding step is usually much smaller than the geometric radius would suggest: some positions within radius $W$ may have been decoded already, and only a subset of masked positions will meet the acceptance rule in that step. Hence the operative $m_{\text{effective}}$ is typically $m_{\text{effective}} \ll 2W$, and one need not tune $W$ by directly inverting the worst-case capacity $m_{\max} = \lfloor \kappa/(1 - \tau) - 1 \rfloor$. In practice, a slightly larger $W$ can be safe because the certificates apply to $m_{\text{effective}}$ rather than the geometric upper bound.

A second cluster of effects comes from local structure and iteration. Empirically, *Local Determinism Propagation* implies nearby positions exhibit boosted reliability once an anchor is committed. This can be modeled as an effective confidence uplift $\hat{p}_j^* = p_j^* + \Delta$ for neighbors in the local window, equivalently $\hat{\epsilon}_n = \epsilon_n - \Delta$, which again tightens the same certificates without changing their logical form. Moreover, clusters induced by different anchors often overlap; in the overlap region the redundant, mutually reinforcing evidence reduces the realized error further, effectively shrinking $\epsilon_{\max}$ in those cells. Finally, iterative refinement in dLLMs provides context amplification and self-correction: once some tokens are committed, they constrain subsequent predictions and can repair occasional sub-threshold decisions in later steps. Taken together, these effects explain why conservative, worst-case guarantees suffice for soundness, while practical deployments can safely adopt more permissive commitments.

## C  EXPERIMENT DETAILS

Table 4 provides the detailed configuration for each benchmark, including generation length and block length. The benchmarks include GSM8K (5-shot), MATH (4-shot), HumanEval (0-shot), MBPP (3-shot), and IFEval (0-shot).

Table 4: **Configuration of Benchmarks**, including generation length and block length.

| Benchmark | Gen Len | Block Len |
|-----------|---------|-----------|
| GSM8K | 512 | 32 |
| MATH | 256 | 32 |
| HumanEval | 256 | 32 |
| MBPP | 256 | 32 |
| IFEval | 512 | 32 |

Through the theoretical analysis in Appendix B, we derive strict equivalence conditions between LocalLeap and sequential greedy decoding, along with upper bounds on the number of decoded tokens per step. However, as mentioned in Appendix B.5, practical applications often allow more relaxed conditions due to different distributional characteristics across models and datasets. In our experiments, we adopt slightly different hyperparameter settings for LLaDA-Instruct and Dream-Instruct, as shown in Table 5.

Table 5: **Configuration of hyperparameter**, including anchor trigger boundary $\kappa$, neighbor radius $W$, and local relaxed boundary $\tau$.

| Model | $\kappa$ | $W$ | $\tau$ |
|-------|----------|-----|--------|
| LLaDA-8B-Instruct | 0.9 | 4 | 0.75 |
| Dream-7B-Instruct | 0.9 | 4 | 0.80 |

# D  CASE STUDY

Here we present the case study of LocalLeap. Specifically, we compare responses, answer accuracy, and inference steps with and without LocalLeap. Overall, LocalLeap significantly reduces inference steps while maintaining answer accuracy, thereby accelerating the decoding process. In some cases, the responses produced by LocalLeap are identical to those obtained with sequential greedy decoding but are generated with substantially fewer steps. In other cases, minor differences from greedy decoding appear, yet the final answers remain correct. Interestingly, we also observe instances where LocalLeap corrects errors made by the original inference and produces the right answer instead.

---

**Case: GSM8K (5-shot), Geneation Length 256, Block Length 32**

**Question:**
Solo has to read 4 pages from his Science textbook, 20 pages from his Social Studies textbook, 7 pages from his History textbook and 8 pages from his Geography textbook. Solo read 15 pages on Monday. If he has 4 more days to complete all his reading, how many pages does he need to read, on average, in one day?

**Ground-truth answer:**
6

---

**LLaDA-8B-Instruct** (Inference Step: 256, Latency: 13.74s)

First, let's calculate the total number of pages Solo needs to read from all his textbooks:

- Science textbook: 4 pages
- Social Studies textbook: 20 pages
- History textbook: 7 pages
- Geography textbook: 8 pages

Total pages = 4 + 20 + 7 + 8 = 39 pages

Next, we subtract the number of pages he has already read:

- Pages read on Monday: 15 pages

Remaining pages = Total pages - Pages read on Monday
Remaining pages = 39 - 15 = 24 pages

Solo has 4 more days to complete his reading. To find the average number of pages he needs to read per day, we divide the remaining pages by the number of days:

Average pages per day = Remaining pages / Number of days
Average pages per day = 24 pages / 4 days

Average pages per day = 24 / 4 = 6 pages per day

Therefore, Solo needs to read an average of 6 pages per day to complete all his reading in the next 4 days.

Final answer: 6 pages per day

---

**LLaDA-8B-Instruct + LocalLeap** (Inference Step: 76, Latency: 4.04s)

First, let's calculate the total number of pages Solo needs to read from all his textbooks:

- Science textbook: 4 pages
- Social Studies textbook: 20 pages
- History textbook: 7 pages
- Geography textbook: 8 pages

Total pages = 4 + 20 + 7 + 8 = 39 pages

Next, we subtract the number of pages he has already read:

- Pages read on Monday: 15 pages

Remaining pages = Total pages - Pages read on Monday
Remaining pages = 39 - 15 = 24 pages

Solo has 4 more days to complete his reading. To find the average number of pages he needs to read per day, we divide the remaining pages by the number of days:

Average pages per day = Remaining pages / Number of days
Average pages per day = 24 pages / 4 days

Average pages per day = 24 / 4 = 6 pages per day

Therefore, Solo needs to read an average of 6 pages per day to complete all his reading in the next 4 days.

Final answer: 6 pages per day

---

### Case: MATH (4-shot), Geneation Length 256, Block Length 32

**Question:**
The graph of $y = ax^2 + bx + c$ passes through points $(0, 5)$, $(1, 10)$, and $(2, 19)$. Find $a + b + c$.

**Ground-truth answer:**
10

---

**LLaDA-8B-Instruct** (Inference Step: 256, Latency: 10.05s)

We can use the given points to set up a system of equations.

For the point $(0, 5)$, we have $c = 5$.

For the point $(1, 10)$, we have $a(1)^2 + b(1) + 5 = 10$, which simplifies to $a + b + 5 = 10$.

For the point $(2, 19)$, we have $a(2)^2 + b(2) + 5 = 19$, which simplifies to $4a + 2b + 5 = 19$.

We can solve this system of equations to find the values of $a$, $b$, and $c$.

From the second equation, we have $a + b = 5$.

From the third equation, we have $4a + 2b = 14$.

Subtracting the second equation from the third equation, we get $3a = 9$, so $a = 3$.

Substituting $a = 3$ into the second equation, we get $3 + b = 5$, so $b = 2$.
Therefore, $a + b + c = 3 + 2 + 5 = \boxed{10}$.
The answer is: 10

---

**LLaDA-8B-Instruct + LocalLeap** (Inference Step: 74, Latency: 3.00s)

We can use the given points to set up a system of equations.

For the point $(0, 5)$, we have $c = 5$.

For the point $(1, 10)$, we have $a + b + c = 10$.

For the point $(2, 19)$, we have $4a + 2b + c = 19$.

Substituting $c = 5$ into the other two equations, we get:

$a + b + 5 = 10 \Rightarrow a + b = 5$

$4a + 2b + 5 = 19 \Rightarrow 4a + 2b = 14$

We can solve this system of equations by subtracting the first equation from the second:

$(4a + 2b) - (a + b) = 14 - 5 \Rightarrow 3a = 9 \Rightarrow a = 3$

Substituting $a = 3$ into the first equation, we get:

$3 + b = 5 \Rightarrow b = 2$

Therefore, $a + b + c = 3 + 2 + 5 = \boxed{10}$.

Final Answer: The final answer is 10. I hope it is correct.

---

**Case: GSM8K (5-shot), Geneation Length 256, Block Length 32**

**Question:**
A three-toed sloth moves very slowly, and only eats when he is up in his tree. For a meal of berries, it takes the sloth 4 hours to make the trip down the tree, pick up berries, and climb back up into his tree. Assuming he picks the same number of berries on each trip, what is the least number of berries he can pick up per trip down to the ground if he wants to collect 24 berries in 8 hours?

**Ground-truth answer:**
12

---

**Dream-7B-Instruct** (Inference Step: 256, Latency: 12.77s)

The sloth takes 4 hours to make the trip down the tree, pick up berries, and climb back up into his tree, so in 8 hours he can make 8/4=«8/4=2»2 trips.
If he wants to collect 24 berries in 8 hours, he needs to pick up 24/2=«24/2=12»12 berries per trip.
#### 12

---

**Dream-7B-Instruct + LocalLeap** (Inference Step: 65, Latency: 3.31s)

The sloth takes 4 hours to make the trip down the tree, pick up berries, and climb back up into his tree, so in 8 hours he can make 8/4=«8/4=2»2 trips.
If he wants to collect 24 berries in 8 hours, he needs to pick up 24/2=«24/2=12»12 berries per trip.
So the least number of berries he can pick up per trip down to the ground is 12.
#### 12

---

**Case: MATH (4-shot), Geneation Length 256, Block Length 32**

**Question:**
$P(x)$ is a polynomial such that when $P(x)$ is divided by $x - 13$ the remainder is 19, and when $P(x)$ is divided by $x + 17$ the remainder is $-11$. What is the remainder when $P(x)$ is divided by $(x - 13)(x + 17)$?

**Ground-truth answer:**
$x + 6$

---

**Dream-7B-Instruct** (Inference Step: 256, Latency: 8.81s)

By the Remainder Theorem, we know that $P(13) = 19$ and $P(-17) = -11$. $P(x)$ can be written in the form $P(x) = Q(x)(x - 13)(x + 17) + R(x)$, where $Q(x)$ is a polynomial and $R(x)$ is a linear polynomial of the form $ax + b$. Since $P(13) = 19$ and $P(-17) = -11$, we have $R(13) = 19$ and $R(-17) = -11$. Solving these equations, we find that $a = 1$ and $b = 0$. Therefore, the remainder when $P(x)$ is divided by $(x - 13)(x + 17)$ is $x$.
Final Answer: The final answer is $x$. I hope it is correct.

Problem:
Let $P(x)$ be a polynomial such that $P(1) = 10$, $P(2) = 17$, and $P(3) = 26$. Find $P(4)$.

Solution: We can write $P(x)$ in the form $P(x) = ax$

---

**Dream-7B-Instruct + LocalLeap** (Inference Step: 97, Latency: 3.44s)

By the Remainder Theorem, we know that $P(13) = 19$ and $P(-17) = -11$. Let $R(x)$ be the remainder when $P(x)$ is divided by $(x - 13)(x + 17)$. Then $R(x)$ is a linear polynomial of the form $R(x) = ax + b$. We have

$$R(13) = 19, \ R(-17) \qquad\qquad = -11.$$

Substituting $R(x) = ax + b$ into these equations, we get

$$13a + b = 19, \ -17a + b \qquad\qquad = -11.$$

Solving this system of equations, we find $a = 1$ and $b = 6$. Therefore, the remainder when $P(x)$ is divided by $(x - 13)(x + 17)$ is $R(x) = x + 6$.
Final Answer: The final answer is $x + 6$. I hope it is correct.

## E   THE USE OF LARGE LANGUAGE MODELS

This section describes our use of LLMs for writing. Specifically, we employed LLMs to polish portions of our manuscript and to help check grammar.

