# OpenReview forum: "Accelerating Diffusion Language Models Inference via Local Determinism Propagation"
_ICLR.cc/2026/Conference — ICLR 2026 Conference Withdrawn Submission_

### Official Review · Reviewer_fF1N · 2025-10-31

**Soundness:** 3
**Presentation:** 4
**Contribution:** 3
**Rating:** 6
**Confidence:** 4

**Summary:**

Identifies problem: dLLMs waste time. They repeatedly refine text that is already correct. Researchers call this "delayed decoding."

The paper proposes LocalLeap. This simple strategy works as a "plug-in" on top of dLLMs. LocalLeap performs two key actions:
- It finds "anchors." that are parts of the text the model is highly confident about.
- It jumps ahead locally. The method assumes areas near these anchors are also safe. It then immediately commits and finalizes several tokens in that neighborhood.

LocalLeap boosts dLLM speed with negligible impact on quality.

**Strengths:**

- Identifies a key pattern in confidence of tokens and presents the problem of delayed decoding. This is very well substantiated empirically.
- Strong throughput improvement (7x) compared to sequential decoding and reduction in number of decoding steps required.
- Quality is almost neutral.
- Training free
- Presents adaptive parallelism while opening venue for further exploration in this direction.

**Weaknesses:**

- Novelty is reasonable and limited.
- Since it is training free, the algorithm is relies on hyperparameters and will be sensitive to them. This can limit whether the same hyperparameters are useful across domains.

**Questions:**

see weakness section

---

### Official Review · Reviewer_3i2X · 2025-11-01

**Soundness:** 3
**Presentation:** 3
**Contribution:** 2
**Rating:** 4
**Confidence:** 3

**Summary:**

This paper presents LocalLeap, an adaptive parallel decoding strategy to accelerate the inference of diffusion large language models. The authors diagnose a "delayed decoding" phenomenon, where conservative sampling schedules lead to redundant refinement iterations. To address this, they propose Local Determinism Propagation and Spatial Consistency Decay to identify high-confidence "anchor" tokens and perform relaxed parallel decoding within their bounded neighborhoods. This method achieves notable inference speedups and significant reduction in decoding steps, with negligible performance degradation.

**Strengths:**

1. The paper diagnoses a key bottleneck in practical dLLM deployment, the analysis of figure 1 provides a solid, empirical foundation for the work.
2. The proposed method is practical and effective, using high-confidence anchors to trigger localized, relaxed decoding is a clever way to leverage the inherent bidirectional nature of dLLMs without requiring retraining.

**Weaknesses:**

1. While the empirical principles of the method are well-supported by heatmaps (figure 2), their theoretical explanation remains somewhat phenomenological. A deeper analysis connecting these observations to the underlying model architecture would strengthen the foundation.
2. Ablation study suggests that the hyper-parameters may not be fully transferable across all dLLMs, potentially requiring per-model tuning for optimal results.
3. The empirical evaluation would be more compelling if it included comparisons with other state-of-the-art, training-free acceleration methods.

**Questions:**

Are there any specific scenarios where the local determinism assumption breaks down, potentially leading to error propagation or performance degradation? It would be better to expand discussion of limitations.

---

### Official Review · Reviewer_bACU · 2025-11-02

**Soundness:** 3
**Presentation:** 2
**Contribution:** 2
**Rating:** 4
**Confidence:** 4

**Summary:**

In this paper, the authors first analyze dLLMs with delayed decoding analysis (to show local determinism propagation and spatial consistency decay) and then propose LocalLeap that (i) detects high-confidence anchors (with a trigger confidence threshold), (ii) opens a local neighborhood of radius W around each anchor, and (iii) early-commits neighbors whose confidence exceeds a relaxed threshold. On LLaDA-8B-Instruct and Dream-7B-Instruct across various benchmarks, LocalLeap reports up to $7\times$ throughput improvement in comparison with baseline dLLMs with minimal accuracy degradation.

**Strengths:**

1. The paper usefully documents dLLM dynamics: (i) confidence-driven local consistency that propagates to neighbors (local determinism), and (ii) to provide visualization on how consistency level drops as distance increase (spatial consistency decay). Both provide intuition for when parallel commitments are safe.

2. The proposed schedule shows clear gains over confidence-based parallel decoding in fast-dLLM, indicating it adds value beyond confidence-only heuristics.

**Weaknesses:**

1. the method adds algorithmic complexity. The method introduces several new knobs (trigger confidence, neighbor radius, relaxed boundary confidence), adding algorithmic complexity.

2. Latency improvements are modest when compared directly to AR baselines, which still benefit from mature KV caching and less FLOPs per step.

3. It would be valuable to analyze which phrase types (e.g. code snippet, short noun phrases) are most reliably parallel-decodable versus long-range/open-ended segments.

**Questions:**

1. Can the authors discuss the overhead of hyper-parameter selection: trigger confidence, neighbor radius, relaxed boundary confidence? Is manual retuning needed per model–benchmark pair, and would there be a online adaptation algorithm for the serving scenario?

2. For the fast-dLLM baseline, did you enable prefix cache only, or a dual KV cache? Please specify cache policy.

3. Could you add a qualitative study on what token clusters (natural language phrases) exhibits better consistency traits, possibly with confidence histograms?

4. Could the authors provide a bit more discussion on what settings (model-task pair) dLLMs’ end-to-end TPS surpass typical AR models?

---

### Official Review · Reviewer_e2gT · 2025-11-04

**Soundness:** 3
**Presentation:** 3
**Contribution:** 3
**Rating:** 4
**Confidence:** 3

**Summary:**

This paper proposes LocalLeap, a training-free adaptive parallel decoding method for diffusion large language models (dLLMs). The core idea is to identify high-confidence "anchor" tokens and expand decoding locally based on neighborhood consistency. The method exploits two observed properties of dLLM decoding: 1. local determinism propagation and 2. spatial consistency decay, to parallelize inference effectively.

**Strengths:**

1. The paper presents a clear motivation and provides insightful empirical observations on token confidence and neighborhood behavior.
2. LocalLeap is simple, generalizable, and training-free, requiring only minor modifications to the decoding loop.
3. Experimental results across multiple datasets and models consistently demonstrate that the proposed method achieves faster inference while maintaining comparable generation quality.
4. The analysis of neighborhood size and threshold parameters is systematic, helping readers understand the quality–efficiency trade-off.

**Weaknesses:**

1. The theoretical analysis of complexity reduction and accuracy trade-offs remains limited. A more formal derivation linking decoding parameters to expected performance would enhance clarity.
2. Hyperparameter tuning (e.g., anchor threshold and neighborhood radius) could be challenging in practice. Automatic or adaptive strategies would improve applicability.
3. The method assumes that token confidence and neighborhood consistency correlate strongly, but the paper does not provide quantitative evidence (e.g., correlation coefficients or empirical distributions) to support this assumption. Without such validation, it is unclear how robust this principle is across different models and domains.


## Minor Issue
1. $V$ in equation 2 is not introduced.

**Questions:**

1. I found Figure 1(a) somewhat unclear. It is not immediately obvious why the consistent predictions are shown in different shades of green. Could the authors clarify what the color variation represents? Additionally, the precise definitions of consistent and inconsistent predictions should be explicitly stated, as they are central to the motivation and interpretation of the figure.
2. The meaning of the dashed lines in Figures 1(b) and 1(c) is not explained. Please specify what these lines indicate to help readers interpret the visualizations correctly.

---

### Note · Authors · 2025-11-21

**Comment:**

We sincerely thank all reviewers for their hard work and constructive feedback. We highly value the insights and suggestions provided, and we will incorporate them to further improve our work in future revisions.

After careful consideration, we have decided to withdraw the paper at this time. We are truly grateful for your time, expertise, and guidance.

**Withdrawal Confirmation:**

I have read and agree with the venue's withdrawal policy on behalf of myself and my co-authors.